# Fucoidan Regulates Starch Digestion: In Vitro and Mechanistic Study

**DOI:** 10.3390/foods11030427

**Published:** 2022-02-01

**Authors:** Hui Si Audrey Koh, Jin Er Leonard Chong, Jun Lu, Weibiao Zhou

**Affiliations:** 1Centre for Life Sciences (CeLS), NUS Graduate School for Integrative Science and Engineering, National University of Singapore, 28 Medical Drive, Singapore 117456, Singapore; fsthsak@nus.edu.sg; 2Department of Food Science & Technology, National University of Singapore, Science Drive 2, Singapore 117542, Singapore; e0442052@u.nus.edu; 3School of Science, Faculty of Health and Environmental Sciences, Auckland University of Technology, 34 St Paul Street, Auckland 1142, New Zealand; 4School of Public Health & Interdisciplinary Studies, Faculty of Health and Environmental Sciences, Auckland University of Technology, 34 St Paul Street, Auckland 1142, New Zealand; 5National University of Singapore (Suzhou) Research Institute, 377 Linquan Street, Suzhou Industrial Park, Suzhou 215123, China

**Keywords:** fucoidan, glycemic index, starch digestion, bread, *Undaria pinnatifida*

## Abstract

Bread is a high glycemic index (GI) food with high amounts of readily digestible carbohydrates. Fucoidan refers to a group of sulfated polysaccharides isolated from brown seaweed that has been gaining traction for its many functional properties, including its ability to inhibit starch hydrolases. In this study, fucoidan was added into bread to lower the glycemic index of bread. Fucoidan fortification at 3.0% reduced the starch digestion rate of baked bread by 21.5% as compared to control baked bread. This translated to a 17.7% reduction in the predicted GI (pGI) with 3.0% of fucoidan. Fucoidan was retained in the bread after baking. Although the in vitro bioavailability of fucoidan was negligible, the in vitro bioaccessibility of fucoidan was high, at 77.1–79.8%. This suggested that although fucoidan may not be absorbed via passive diffusion, there is potential for the fucoidan to be absorbed via other modes of absorption. Thus, there is a potential for the use of fucoidan as a functional ingredient in bread to reduce the glycemic potential of bread.

## 1. Introduction

Bread is a staple food that is widely consumed in many parts of the world. However, bread contains a large amount of readily digestible carbohydrates and is considered to be a carbohydrate-rich food item with high glycemic index (GI). The glycemic index (GI) of food is defined as the blood-glucose-raising effect of digestible carbohydrates in a given food [1]. It has been reported that diets high in GI are associated with a 10–33% increased risk of Type 2 diabetes mellitus (T2DM) [2]. On the other hand, diets low in GI are associated with lower risks of glycemia, dyslipidemia, and cardiovascular issues [3]. As such, there is a growing interest in exploring ways to lower the GI of bread. 

One way to lower the GI of bread is through reformulation and fortification with functional ingredients [4,5]. Sulphated polysaccharides isolated from various species of seaweed were reported to inhibit starch digestive enzymes such as α-amylase and α-glucosidase [6]. Fucoidan refers to a group of sulfated polysaccharides isolated from brown seaweed species, with numerous bioactive properties [7]. Fucoidan isolated from different seaweed species exhibits different structures, and hence bioactivities [7]. In particular, fucoidan isolated from *Undaria pinnatifida,* with its unique backbone structure of alternating fucose–galactose units, has been reported to exhibit unique functional properties, including anticancer and antioxidant capabilities [8]. It also was reported in our previous study that fucoidan from *U. pinnatifida* inhibited the starch-digesting enzymes α-amylase, α-glucosidase, and amyloglucosidase [9]. Hence, there is potential to use fucoidan from *U. pinnatifida* as a functional ingredient in bread to lower starch digestibility. The feasibility of incorporating fucoidan into baked bread also was demonstrated in our earlier study, in which the fortification of fucoidan resulted in the production of baked bread with a larger specific volume and a softer crumb texture [10]. The antioxidant and anticancer activities of fucoidan were also retained after the baking process, and the antioxidant activity of fucoidan was reported to be enhanced after baking [10]. However, one major concern with regard to the fortification of bread with functional ingredients is the bioavailability and bioaccessibility of the added ingredients. Hence, there is the need to determine if fucoidan can be bioavailable and/or bioaccessible after the digestion process for fucoidan to produce the desirable bioactivities, such as anticancer and antioxidant activity. We hypothesized that the incorporation of fucoidan into a bread formula would reduce the in vitro starch digestibility and glycemic potential of baked bread.

This study aimed to investigate the digestion profile of baked bread fortified with fucoidan from *U. pinnatifida*, in which the glycamic lowering effect of fucoidan on baked bread was determined. This was one of the first studies to investigate the ability of fucoidan to retain its bioactivity—specifically, inhibiting starch-digesting enzymes—after incorporation into a food matrix.

## 2. Materials and Methods

### 2.1. Chemical and Samples

Fucoidan isolated from the brown seaweed species *U. pinnatifida* was obtained from Auckland, New Zealand [11]. We reported in our previous study that fucoidan from *U. pinnatifida* was composed of a unique backbone structure with alternating fucose and galactose units, and sulfation at both the C2 and C4 positions, as shown in Figure 1 [8]. This unique structure of fucoidan has been demonstrated to exhibit strong inhibitory activities against starch-digesting enzymes such as amylase, amyloglucosidase, and glucosidase, as reported in our previous work [9]. The binding of fucoidan to these enzymes was proposed to be mediated via electrostatic interactions between the negatively charged sulfate groups of fucoidan and the positive charges of the enzyme molecules [9].

Bread flour (Prima Limited, Singapore), salt (Pagoda, Siem Trading, Singapore), sugar (NTUC Fairprice, Singapore), dry instant yeast (*Saccharomyces cerevisiae*, S.I. Lesaffre, Lille, France), and shortening (Bake King, Gin Hin Lee, Singapore) were purchased from Fairprice, Singapore. Alpha amylase from porcine pancreas (A3176), amyloglucosidase from *Aspergillus niger* (A7095), dialysis tubing cellulose membrane (D9777), pancreatin from porcine pancreas (P7545), pepsin from porcine gastric mucosa (P7000), and porcine bile extract (B8631) were obtained from Sigma-Aldrich (St. Louis, MO, USA). The L-fucose assay test kit (K-FUCOSE), available carbohydrates/dietary fiber assay kit (K-ACHDF), and D-fructose/D-glucose assay kit (K-FRUGL) were obtained from Megazyme (Bray, Dublin, Ireland). 

### 2.2. Preparation of Bread Samples

The bread samples were prepared based on 330 g of dough: 200 g bread flour, 122 g water, 8 g sugar, 6 g shortening, 2.4 g salt and 2.0 g dry instant yeast. Fucoidan isolated from *U. pinnatifida* were incorporated into the dough and bread samples at 0.0%, 1.5%, 2.0%, 2.5%, and 3.0% w/w flour weight. The dough and bread samples were prepared according to the no-time dough method by Ananingsih, Gao, and Zhou [12] with slight modifications that allowed the bread to be made in a shorter time by using -energy mixing to speed up gluten development. Briefly, the ingredients were mixed using a mixer (Model 5KPM50, Kitchen Aid, Troy, OH, USA) at 45 rpm for 1 min and at 100 rpm for 5 min to produce the bread dough. The dough sample was allowed to rest for 10 min before being divided and molded in individual dough pieces weighing 55 ± 1 g. The dough pieces were proofed at 37 °C and 85% relative humidity for 70 min before baking at 200 °C for 10 min. After baking, the baked bread samples were left to cool to room temperature, and the crust of the bread was removed. The breadcrumbs were blended and sieved using a 40-micron sieve to obtain the bread samples used for the subsequent in-vitro digestion experiment.

### 2.3. Quantification of Fucoidan

#### 2.3.1. Fucoidan Extraction

The baked bread samples were freeze-dried and milled using a blender to 40 μm, and passed through a sieve before analysis, according to the method described by Koh et al. [10]. Briefly, bread powder (0.5 g) was extracted with 5 mL of deionized water via sonication (Elmasonic S60H, Elma Schmidbauer GmbH, Singen, Germany) for 15 min and shaking for 10 min. The sample was centrifuged, and the supernatant was decanted into a 50 mL volumetric flask. The extraction cycle was repeated for a total of 6 cycles, and the supernatant collected was combined and topped up to 50 mL with deionized water to form the extracts.

#### 2.3.2. Quantification of Fucoidan

Fucoidan in the extracted samples was quantified using an L-fucose commercial assay kit (Megazyme, Bray, Dublin, Ireland), according to the method described by Koh et al. [9]. Briefly, the fucoidan extract (5 mL) was acid-hydrolyzed (1 mL 4 M HCl) at 100 °C for 60 min, neutralized with 4 M NaOH upon cooling, and quantified using the L-fucose commercial assay kit.

### 2.4. In Vitro Digestibility Study

The in vitro digestibility study was performed according to the standardized static in vitro digestion protocol developed by Minekus et al. [13] with slight modification. The in vitro digestion process consisted of three phases—oral, gastric, and intestinal—and was conducted in a 37 °C water bath (MX-CA21E, Polyscience, Niles, IL, USA) equipped with a magnetic stirrer board (MIXdrive 15, 2mag, Munich, Germany).

#### 2.4.1. Oral Phase

The oral phase was initiated by mixing 5 g of breadcrumbs with 4 mL of α-amylase in SSF buffer (75 U/mL in the final mixture) and 1 mL of 7.5 mM CaCl_2_ solution. The mixture was vortex for 20 s and incubated in the 37 °C water bath for 2 min with constant stirring at 350 rpm.

#### 2.4.2. Gastric Phase

The resulting sample obtained after the oral phase was mixed with 8 mL of pepsin in SGF buffer (2000 U/mL in the final mixture), 5 μL of 0.3 M CaCl_2_, and 1.345 mL of deionized water. The mixture was then adjusted to pH 3.0 with 0.65 mL of 1 M HCl solution. The resulting mixture was incubated in the 37 °C water bath with constant stirring at 450 rpm for 2 h.

#### 2.4.3. Intestinal Phase

The chyme obtained after the gastric phase was mixed with 16 mL of pancreatin in SIF buffer (100 U/mL for trypsin activity in the final mixture), 80 μL of amyloglucosidase (21 U/mL in the final mixture), 0.1768 g of bile, 3.23 mL of deionized water, and 40 μL of 0.3 M CaCl_2_ solution. The pH of the resulting mixture was adjusted to pH 7.0 using 425 μL of 0.2 M NaOH solution. The digestion mixture was transferred into a dialysis tube (with a cut-off size of 14 kDa) and dialyzed in 200 mL of PBS buffer at 37 °C with constant stirring at 450 rpm for 7 h.

### 2.5. Reducing Sugar Release

Throughout the intestinal phase, aliquots of 0.5 mL of dialysate were withdrawn at 0, 5, 10, 15, 30, 45, 60, 75, 90, 105, 120, 150, 180, 210, 240, 300, 360, and 420 min, and stored in 2 mL centrifuge tubes for subsequent analysis. The amount of reducing sugar released into the dialysate at each time interval was determined using the D-fructose/D-glucose assay kit from Megazyme (K-FRUGL, Megazyme, Bray, Ireland).

### 2.6. Mathematical Modelling

The digestion curve obtained from the in vitro digestion system followed a first-order reaction. The reducing sugar release data was modelled using Equation (1) as proposed by Goñi, Garcia-Alonso, and Saura-Calixto [14] in order to determine the extent by which the different levels of fucoidan fortification affected the digestibility of the bread samples:(1)$$C_{t}=C_{\infty }\left(1-e^{-kt}\right)$$
where *C_t_* refers to the concentration of reducing sugar released at time *t*, *C_∞_* refers to the equilibrium concentration of reducing sugar released, *k* refers to the rate constant of starch digestion (min^−1^), and *t* refers to time (min).

The digestion curves were modelled using GraphPad Prism graphing software version 7.0a (GraphPad Software Inc., San Diego, CA, USA) to obtain the *k* and *C*_∞_ values.

### 2.7. Prediction of GI and GL (Glycemic Load)

#### 2.7.1. Total Available Carbohydrate (TAC)

The total available carbohydrate (TAC) of each bread sample was measured using the Available Carbohydrate/Dietary Fiber assay kit from Megazyme (K-ACHO, Megazyme, Dublin, Ireland), based on the study by Lin, Teo, Leong, and Zhou [15]. The results obtained were expressed in terms of mg of TAC per 5 g of fresh bread for all the different bread samples.

#### 2.7.2. Estimation of GI and GL

Experimental data obtained from the in vitro digestion of bread samples were used to predict the GI and GL of the bread samples according to the method by Wolter, Hager, Zannini, and Arent [16]. The amount of reducing sugar released was expressed in terms of grams of reducing sugars released per 100 g of TAC of fresh bread, and plotted against the digestion time (min) for 180 min. Although the in vitro digestion experiment was allowed to proceed for a total of 7 h, only the first 180 min of data were used in the estimation of GI and GL, as the transit of food through the small intestine takes approximately 3 h in the human body [13]. The area under the curve (AUC, g per 100 g TAC per min) was calculated using GraphPad Prism graphing software version 7.0a (GraphPad Software Inc., San Diego, CA, USA). The hydrolysis index (HI) was then determined from the AUC according to Equation (2):(2)$$HI=\left(\frac{AUC_{sample}}{AUC_{control\;wheat\;bread}}\right)\times 100$$

The predicted GI (pGI) values of the bread samples were computed using the equation pGI = 0.549 HI + 39.71, developed by Goñi, Garcia-Alonso, and Saura-Calixto [14]. In this case, the reference food was white bread, and the pGI calculated was termed pGI_bread_ (where the GI value of white bread = 100). In order to estimate the GI of the bread samples using glucose as the reference (pGI_glucose_, where the GI of glucose = 100), the pGI_bread_ of each sample was multiplied by a factor of 0.7 according to Wolever et al. (2008) [17]. The predicted GL of the bread samples (pGL) was computed based on a 50 g bread sample according to Equation (3):(3)$$pGL=\frac{pGI_{glucose}\times TAC}{100}$$

### 2.8. Recovery of Fucoidan after Digestion

#### 2.8.1. Sample Treatment

At the end of digestion process, the digesta, dialysate, and sediments were collected for further analysis. Dialysate refers to the solution outside the dialysis tube. The intestinal chyme was centrifuged, and the digesta (supernatant) and sediments (pellets) were collected separately.

The dialysate was concentrated using a vacuum rotary evaporator (N-1200A, Eyela, Tokyo, Japan) to less than 50 mL. The resulting solution was topped up to 50 mL using deionized water. The digesta was transferred into a 50 mL volumetric flask and topped up to 50 mL using deionized water.

#### 2.8.2. Fucoidan Extraction

The sediments were placed in a 15 mL centrifuge tube and mixed with 5 mL of deionized water. The mixture was vortexed and sonicated for 15 min (Elmasonic S60H, Elma Schmidbauer GmbH, Singen, Germany). The resulting samples were then shaken using an orbital shaker at 300 rpm for 10 min. After shaking, the tubes were allowed to stand for 5 min and centrifuged at 514× *g* for 2 min before decanting the supernatant into a 50 mL volumetric flask. The entire extraction cycle was repeated using the residue for another 5 cycles, and the supernatant obtained was combined and topped up to 50 mL using deionized water. The resulting extracts were passed through 0.22 μm nylon syringe filters (Thermo Fisher Scientific Inc, Waltham, MA, USA).

#### 2.8.3. Fucoidan Quantification

The amount of fucoidan in the dialysate, digesta, and sediments was quantified using the L-fucose assay kit by Megazyme (K-FUCOSE, 08/16, Megazyme, Dublin, Ireland) according to the process given in Section 2.3.2, using 1 mL of extract instead of 5 mL.

#### 2.8.4. Determination of Potential Bioaccessibility and Bioavailability

The potential bioaccessibility of fucoidan from the in vitro digestion system was defined as the total amount of fucoidan released from the bread matrix during digestion and released into the digestion solution [15]. The potential bioavailability of fucoidan from the in vitro digestion system was defined as the total amount of fucoidan released from the bread matrix during digestion and passed through the dialysis tube into the surrounding PBS buffer solution, mimicking the amount of fucoidan that can pass through intestinal walls and be absorbed in vivo [15].

The in vitro bioaccessibility and bioavailability of fucoidan were determined according to Equations (4) and (5), respectively:(4)$$F_{AC}\left(\%\right)=\frac{F_{digesta}+F_{dialysate}}{F_{total}}\times 100$$
(5)$$F_{AV}\left(\%\right)=\frac{F_{digesta}}{F_{total}}\times 100$$
where *F_AC_* refers to the in vitro bioaccessibility of fucoidan; *F_digesta_* refers to the amount of fucoidan in the digesta fraction; *F_dialysate_* refers to the amount of fucoidan in the dialysate fraction; *F_total_* refers to the total amount of fucoidan in the digesta, dialysate, and sediment fraction; and *F_AV_* refers to the in vitro bioavailability of fucoidan.

A separate in vitro digestion experiment (3% matched) was conducted using control baked bread spiked with fucoidan at a level that was equivalent to the amount of fucoidan left in the 3% bread after baking. This was to investigate the effect of baking on fucoidan in terms of its ability to retard starch digestion.

### 2.9. Statistical Analysis

All analyses were carried out in triplicates, with two or three repeats in each replicate. The results were processed and expressed as the mean values along with their individual standard deviations. Statistical analysis was performed using XLSTAT 2016 software (Addinsoft, New York, NY, USA). One-way ANOVA (IBM Corporation, New York, NY, USA) was used to carry out statistical analysis before a post hoc analysis was carried out using Tukey’s range test. 

## 3. Results and Discussion

### 3.1. Recovery of Fucoidan after Baking

The recovery rate of fucoidan in bread after baking is shown in Figure 2. The recovery rate of fucoidan was 70.3–82.1% after baking. This suggested a loss of approximately 18–30% of fucoidan during the baking process. The decrease in fucoidan could be accounted for by yeast fermentation during proofing [18]. Yoon, Mukerjea, and Robyt [18] and Wilkinson [19] previously reported the ability of baker’s yeast to ferment galactose, a major monosaccharide present in fucoidan from *U. pinnatifida*. Although the backbone structure of fucoidan is composed of galactose and fucose subunits, it is unlikely that yeast is able to use galactose in polysaccharide form as a substrate for fermentation. However, it was observed in our previous study that the total amount of gas produced in the fucoidan dough samples was significantly higher than that of the control dough samples [10]. This suggests that yeast might have utilized the low-molecular-weight fraction of fucoidan as a substrate to produce gas.

Moreover, it was reported that fucoidan may be susceptible to degradation at high temperatures [20]. Since the production of bread involves baking at high temperatures of 200 °C, it is likely that a fraction of the fucoidan polysaccharides was broken down into smaller units or their monosaccharide components. Under high-temperature conditions and in the presence of wheat flour proteins in the dough, it is possible that the low-molecular-weight fucoidan fragments and their monosaccharide subunits reacted with amino acids in a Maillard reaction to produce Maillard reaction productions [21]. This may also account for the loss in fucoidan content after the baking process.

There was also a significant difference in the recovery rate of fucoidan among the samples, and this difference can be attributed to the different amount of fucoidan utilized by yeast for proofing. Alternatively, this difference can also be attributed to the variability in the breakdown of fucoidan during baking. Nonetheless, a good recovery of approximately 70.3–82.1% of fucoidan was found in this study.

### 3.2. In Vitro Starch Digestibility of Bread

Figure 3 shows the digestion curve of bread samples throughout the 7 h of the intestinal phase. From Figure 3, it can be observed that the addition of fucoidan slowed down starch digestion in the baked bread samples, where the concentration of reducing sugar (RS) in the fucoidan-fortified bread was lower than that of the control baked bread. At the end of 105 min of digestion, the concentration of RS released by the fucoidan-fortified bread was 23–34%, which was significantly lower than that by the control baked bread. This suggested that fucoidan was able to significantly reduce starch digestion in bread as early as 105 min into the intestinal phase.

It was reported in the literature that the typical duration required for intestinal chyme to transit through the small intestine is 3–4 h [13]. In our study, at the end of 4 h of the intestinal phase (i.e., t = 240 min), the concentration of RS in the 2.0% and 2.5% fucoidan-fortified bread was ~15%, significantly lower than that by the control bread, while the concentration of RS in the 3.0% fucoidan-fortified bread was 17%, significantly lower than that by the control bread. This suggested a dose-dependent inhibition of starch digestion in baked bread by fucoidan, where the concentration of RS released was reduced (i.e., the inhibition of starch was enhanced) with an increasing concentration of fucoidan.

One possible explanation for the observed phenomenon is the ability of fucoidan to inhibit a wide range of starch-digesting enzymes. Fucoidan from various species of seaweeds has been reported to be an inhibitor of starch hydrolases [6]. In our previous study, fucoidan from *U. pinnatifida* was shown to be an uncompetitive inhibitor of α-amylase and amyloglucosidase [9]. This meant that fucoidan preferentially bound to the enzyme–substrate complex instead of binding to the enzyme itself, a similar inhibition mechanism to that of acarbose on α-amylase [22]. It was suggested that fucoidan had a poorer affinity to the active site of α-amylase than the substrate. However, upon the binding of substrate to α-amylase, it activated a secondary binding site on α-amylase to which fucoidan had a high affinity [22]. Binding of fucoidan to the secondary binding site on the enzyme–substrate complex in turn led to the inactivation of α-amylase.

It is proposed that fucoidan inhibited amyloglucosidase in a similar mechanism, where fucoidan preferentially bound to a secondary binding site on amyloglucosidase that was activated after the substrate had bound to amyloglucosidase. This binding between fucoidan and the secondary binding site of amyloglucosidase could be facilitated by the electrostatic interactions between the negatively charged sulfate groups of fucoidan and the enzyme’s secondary binding site [23].

The inhibition of starch digestion in bread by fucoidan might also be due to the inhibitory activity of fucoidan on α-glucosidase inhibition. The presence of a hydrogen ion at its catalytic site is required for α-glucosidase to hydrolyze α(1–4) glucosidic bonds [6]. However, in the presence of fucoidan, it was proposed that fucoidan could scavenge the hydrogen ion at the catalytic site of α-glucosidase, and thus inhibit its activity [24,25]. Alternatively, fucoidan might also inhibit α-glucosidase by competing with the substrate for the active site of α-glucosidase, similar to the mode of inhibition of acarbose [26]. Similarly, the binding of fucoidan to the active site of α-glucosidase might be modulated by the electrostatic forces of attraction between the negatively charged fucoidan and the active site of α-glucosidase [27,28,29].

Another mechanism is that fucoidan, being a polysaccharide, slowed down the diffusion of glucose from the enzyme’s active site by increasing the viscosity of the intestinal chyme [30,31,32].

### 3.3. Mathematical Modelling

Comparing the concentrations of RS released at individual time points alone is insufficient to support the hypothesis that fucoidan fortification reduced starch digestion in baked bread significantly. Therefore, mathematical modelling was necessary to validate the hypothesis by providing digestion rate constants for all the bread samples. The equilibrium concentration for all models was set to the theoretical equilibrium for all bread samples at 12.16 mg/mL. As shown in Table 1, the digestion rate constant, *k*, decreased significantly with an increasing concentration of fucoidan fortification. The digestion rate constant gave a quantifiable indication of the digestion rate. As such, the lower the *k* value, the slower the rate of starch digestion in the bread samples. The maximum reduction in the digestion rate of 21.5% was observed at the highest fortification level of 3%. This further supported the hypothesis that fucoidan fortification reduced starch digestion in baked bread. This was in good agreement with the literature reports in which anthocyanin-rich black rice extract fortification at 4% and quercetin fortification at 6% reduced the digestion rate constant of baked bread by 18.3% and 20.5%, respectively [15,33].

In order to evaluate the model performance, the modelled digestion data of all the bread samples were plotted against the experimental data, as shown in Figure 4. It can be seen that the data points were evenly distributed along the 45° dotted line, indicating a good agreement between the experimental and modelled data. Furthermore, the low RMSE values in Table 1 (RMSE < 0.5) indicated a relatively good model with small differences between the experimental and modelled data, since RMSE measures the absolute fit of the model to the data. The high R^2^ values shown in Table 1 (R^2^ > 0.9), where R^2^ measured how close the data were to the fitted line, further confirmed the appropriateness of the model to be used for describing the digestion profile of the bread samples.

### 3.4. Prediction of GI and GL

The total available carbohydrate (TAC)-predicted GI and GL values for all the bread samples are tabulated in Table 1. It was observed that fucoidan fortification did not result in a significant difference in the TAC content of all five bread samples. However, the pGI_bread_ and pGI_glucose_ values of all four fucoidan-fortified bread samples were significantly lower than that of the control bread. This further supported the hypothesis that fucoidan slowed down starch digestion in bread. Fucoidan is a known inhibitor of the starch-digesting enzymes α-amylase, α-glucosidase, and amyloglucosidase; therefore, in the presence of fucoidan, the activity of these enzymes was reduced, thereby delaying starch digestion and lowering the pGI of the bread samples. Furthermore, fucoidan as a polysaccharide increased the viscosity of the intestinal chyme [30,31]. This further delayed the association of the substrate with enzymes, and the dissociation of glucose/reducing sugar from the enzymes, thereby leading to a lower GI.

It was observed that the maximum reduction in pGI_bread_ and pGI_glucose_ occurred at the highest level of fucoidan fortification at 3%, where pGI_bread_ and pGI_glucose_ were both lowered by 17.7%. Likewise, the maximum reduction in pGL was observed at the highest level of fucoidan fortification, where pGL was reduced by 19.4%. Glycemic load (GL) is a better indication of glycemic response of a food item, as it takes into account both the amount of total carbohydrates in a portion of food, as well as how quickly blood glucose level rises [34]. A linear dose-dependent reduction in pGI and pGL was also observed with increasing concentrations of fucoidan (linear correlation regression R^2^ = 0.984; *p* < 0.05). 

Typically, the GI_glucose_ of white wheat bread is 100, while that of wholemeal bread is 74 [35]. In our study, it was demonstrated that the pGI_glucose_ of white bread could be lowered to 57.5 ± 4.5 with fucoidan fortification of 3%. This was significantly lower than that of whole wheat bread and many other carbohydrate-rich food products, including brown rice (GI = 68 ± 4) [35]. Thus, fucoidan-fortified bread may be a “healthier” alternative to many of the carbohydrate food items that are widely consumed as staple foods in many parts of the world. However, it was noteworthy that the pGI values obtained from this in vitro study were only an estimation of GI values of food obtained via an in vivo study. 

In the literature, it was shown that quercetin fortification (1.5–6%), anthocyanin-rich black tea extract fortification (1–4%), baobab fruit extract fortification (1.88–3.33%), and green tea extract fortification (0.4–2%) all successfully reduced the starch digestion of bread products [15,33,36]. This implied that all the above-mentioned ingredients had the potential to be exploited in bread products to reduce in vivo glycemic response. However, it also was reported in the literature that baobab fruit extract fortification (1.88%) and green tea extract fortification (0.4%) did not produce a significant reduction in glycemic response or hunger when administered to a group of 13 healthy volunteers [36]. These contradictory results between in vitro and in vivo glycemic responses highlight the need for further human studies to further confirm the effect of fucoidan fortification on the glycemic response of baked bread. 

### 3.5. Potential Bioaccessibility and Bioavailability of Fucoidan after Digestion

The amount of a particular nutrient present in food that is eaten can be very different from the amount of the nutrient present in the intestinal lumen that is released from the food matrix after digestion [37,38]. Likewise, the amount of a nutrient present in the intestinal lumen after digestion of a particular food item may be very different from the amount of the nutrient that is available for absorption by various cells and tissues in the circulation system [37,38]. Therefore, it is necessary to determine the bioaccessibility and bioavailability of any nutrient when evaluating the nutritive value of the nutrient [37,39].

Bioavailability is defined as the amount of a particular nutrient that is absorbed and available for physiological function after digestion (i.e., the potential bioavailability is the amount of fucoidan in dialysate), while bioaccessibility is defined as the amount of a particular nutrient that can be potentially absorbed in the intestinal lumen (i.e., the potential bioaccessibility includes fucoidan in both the dialysate and digesta) [15,39]. 

The amounts of fucoidan present in the dialysate, digesta, and sediment after in vitro digestion are shown in Table 2. It was observed that a negligible quantity of fucoidan was detected in the dialysate after the in vitro digestion process. This suggested that very little or no fucoidan could pass through the dialysis tube into the surrounding PBS buffer. However, it was noteworthy that the overall recovery of fucoidan was high (89.2–93.8%). This implied that most of the fucoidan was present in the digesta and sediment fraction of the intestinal chyme, as shown in Table 2.

The overall in vitro bioaccessibility and bioavailability of fucoidan in all the bread samples are tabulated in Table 2. From the results, it was concluded that fucoidan was not bioavailable when added into a bread matrix. Given that the potential bioavailability refers to the amount of fucoidan that is absorbed in the intestinal lumen and available for potential physiological function, the negligible quantity of fucoidan detected in the dialysate implied that fucoidan was not bioavailable when placed in a bread matrix. This negligible bioavailability of fucoidan in the fucoidan-fortified bread can be attributed to the large molecular size of the fucoidan used. The fucoidan added into the bread formulation had a molecular cut-off of 300 kDa. However, the dialysis tube used had a pore size of 14 kDa to simulate the small intestine. Therefore, despite the possibility of fucoidan to be broken down into smaller molecules during baking, it is unlikely that the fucoidan would be broken down in small enough molecules to pass through the dialysis tube via passive diffusion into the dialysate fraction. This was in good agreement with the study by Zhao et al. [40], in which the in vivo bioavailability of low-molecular-weight fucoidan fragments (7.6 kDa) was higher than that of the high-molecular-weight fucoidan fragments (35 kDa) when orally administered in rats. It is likely that the absorption for such a large-molecular-weight polysaccharide occurred via other mechanisms. In a separate study, Nagamine, Hayakawa, Nakazato, and Iha [41] reported the absorption of fucoidan via the Caco-2 cell line.

Moreover, it has been suggested that fucoidan inhibits starch digestion via binding to the enzyme or the enzyme–substrate complex. This further reduced the possibility of the fucoidan–enzyme complex to pass through the pores of the dialysis tube (the molecular size of α-amylase is 51–54 kDa, Sigma Aldrich, St. Louis, MO, USA).

As such, the potential in vitro bioavailability of fucoidan in the bread was negligible when the only mode of transport available in the in vitro digestion system was passive transport.

The potential bioaccessibility of fucoidan after digestion of the fucoidan-fortified bread was 77.1–79.8% of the total amount of fucoidan retained in the bread after baking. The potential bioaccessibility of fucoidan was defined as the amount of fucoidan in the digesta that could be potentially absorbed. As such, the potential bioaccessibility was independent of the mode of transport across the dialysis tube, and was only dependent on digestion and release from the food matrix [37,39]. The relatively high bioaccessibility of fucoidan after digestion suggested that most of the fucoidan was released into the liquid portion of the intestinal chyme that could potentially be absorbed via other means, such as active transport [41]. In other words, fucoidan could not be ruled out from potentially being transported via the systematic circulation to target cells and tissues to elicit its bioactivities. Moreover, the retention of fucoidan within the intestine may not necessarily imply that the fucoidan was not viable as a functional ingredient. It was demonstrated in this study that fucoidan could reduce in vitro starch digestion in a high-carbohydrate food item such as bread and reduce the pGI and GL of the food system. In addition, the retention of fucoidan in the intestine may allow it to serve as both an antioxidant and anticancer agent in the intestine. Furthermore, carbohydrates that cannot be digested by the body’s digestive system have the potential to serve as prebiotics [42]. Since fucoidan is a polysaccharide, there is potential for it to serve as prebiotic to support the growth of probiotics in the gut [43]. This was supported by a study by Zaporozhet et al. [43] in which fucoidan enhanced the growth and survival of *Bifidobacterium lactis* probiotics in an in vitro system.

### 3.6. Effect of Bread Matrix on Fucoidan during Digestion

To investigate the effect of bread matrix on fucoidan during and after digestion, control baked bread samples were spiked with an equivalent amount of fucoidan retained in the 3% fucoidan-fortified bread after baking. The digestion profiles of the two bread samples are presented in Figure 5. It was observed that at the end of the 7 h digestion, the 3% matched bread sample had a higher amount of RS in the dialysate as compared to the 3% fucoidan-fortified bread samples. This suggested that that fortifying fucoidan into bread before baking would be more effective in inhibiting starch digestion.

In order to validate the above hypothesis, the digestion curves were modelled to generate the values of the digestion rate constant, *k*, which are shown in Figure 5. It was observed that the 3% matched bread had a significantly higher *k* value than the 3% fucoidan-fortified bread, indicating a significantly higher rate of starch digestion in the 3% matched bread sample despite containing the same initial amount of fucoidan before digestion. This difference can be attributed to the differences in molecular size of fucoidan present in both bread samples. Fucoidan is susceptible to degradation at high temperatures [20]. Therefore, it is likely that during the baking process (at 200 °C), fucoidan molecules in the bread samples were degraded into smaller-molecular-weight fractions. It was reported in the literature that the starch hydrolase inhibition activity of fucoidan depended on the molecular weight of the polysaccharide [6]. Kim, Rioux, and Turgeon [6] compared the α-amylase inhibition activity of fucoidan from *F. vesiculosus* (higher molecular weight) and fucoidan from *A. nodosum* (lower molecular weight), and concluded that fucoidan from *A. nodosum* was a stronger inhibitor of α-amylase. The authors attributed the difference in α-amylase inhibition activity to the different molecular weights [6,20]. It was proposed that higher-molecular-weight fucoidan possessed a structural conformation that was unfavorable to interacting with the enzyme [6]. As such, it is plausible to propose that the 3% fucoidan-fortified bread contained lower-molecular-weight fucoidan that was more effective in inhibiting starch digestion than the 3% matched fucoidan of a higher molecular weight.

## 4. Conclusions

The results of this study highlighted the potential of fucoidan to be used as a functional ingredient in the reformulation of baked bread in order to lower the glycemic potential of white bread. Compared to the control bread, the amount of RS released at the end of 105 min of intestinal phase digestion was 23–34% lower in the fucoidan-fortified baked bread samples at 1.5%, 2%, 2.5%, and 3%. Fucoidan-fortified baked bread at 3% significantly reduced the starch digestion rate and glycemic potential (pGL) by 21.5% and 19.4%, respectively, over the control bread. The resultant pGI_glucose_ of the 3% fucoidan-fortified baked bread at 57.6 ± 4.5 was lower than most high-carbohydrate foods, including brown rice, allowing it to serve as an “healthier” food option for consumers. It was also revealed that fucoidan had negligible potential bioavailability in the in vitro digestion model due to its large molecular weight. However, fucoidan had high potential bioaccessibility, suggesting its potential to be absorbed in vivo via active transport and/or endocytosis. Lastly, the matrix effect of bread on the starch hydrolase inhibition activity of fucoidan was investigated. The 3% fucoidan-fortified bread had a significantly lower starch digestion rate than the 3% matched fucoidan bread. One possible explanation is that fucoidan fortification into bread before baking allowed the large-molecular-weight fucoidan to be degraded into lower-molecular-weight fucoidan units. The smaller fucoidan molecule possessed less structural hindrance to interacting with starch hydrolyzing enzymes such as α-amylase, and was thus more effective in reducing the starch digestion rate.

## Figures and Tables

**Figure 1 foods-11-00427-f001:**
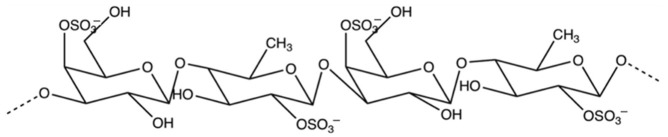
Structure of fucoidan isolated from *U. pinnatifida*.

**Figure 2 foods-11-00427-f002:**
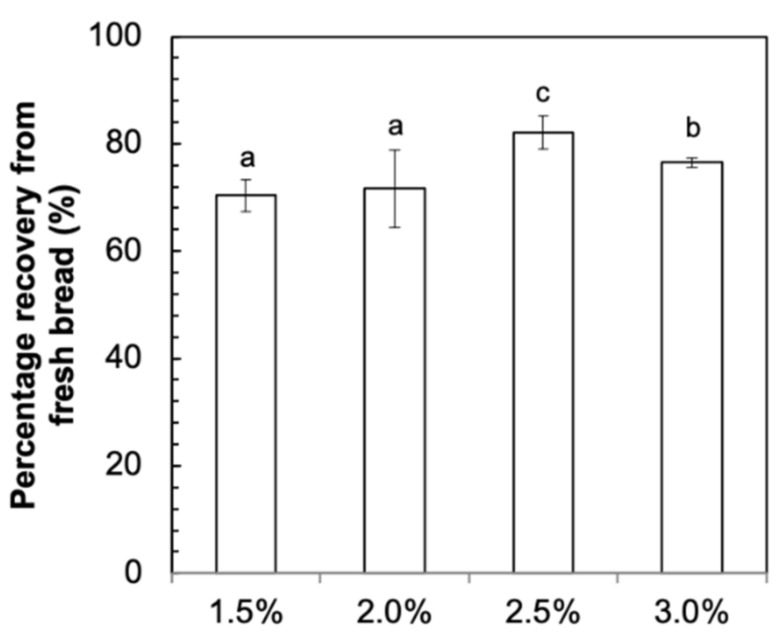
Recovery of fucoidan in bread after baking; ^a,b,c^ Values are presented as mean with standard deviation (*n* = 6). Mean values with different superscript lowercase letters were statistically different (*p* < 0.05) across the different samples.

**Figure 3 foods-11-00427-f003:**
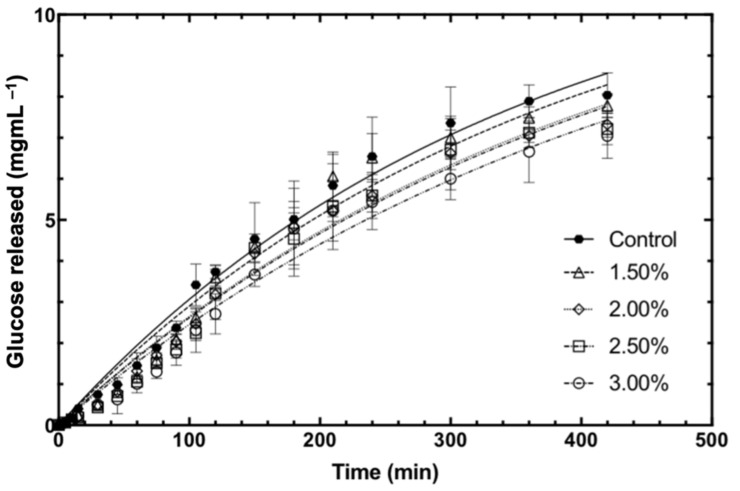
The digestion trajectories of 0%, 1.5%, 2%, 2.5%, and 3% fucoidan-fortified bread and the corresponding developed mathematical models (lines).

**Figure 4 foods-11-00427-f004:**
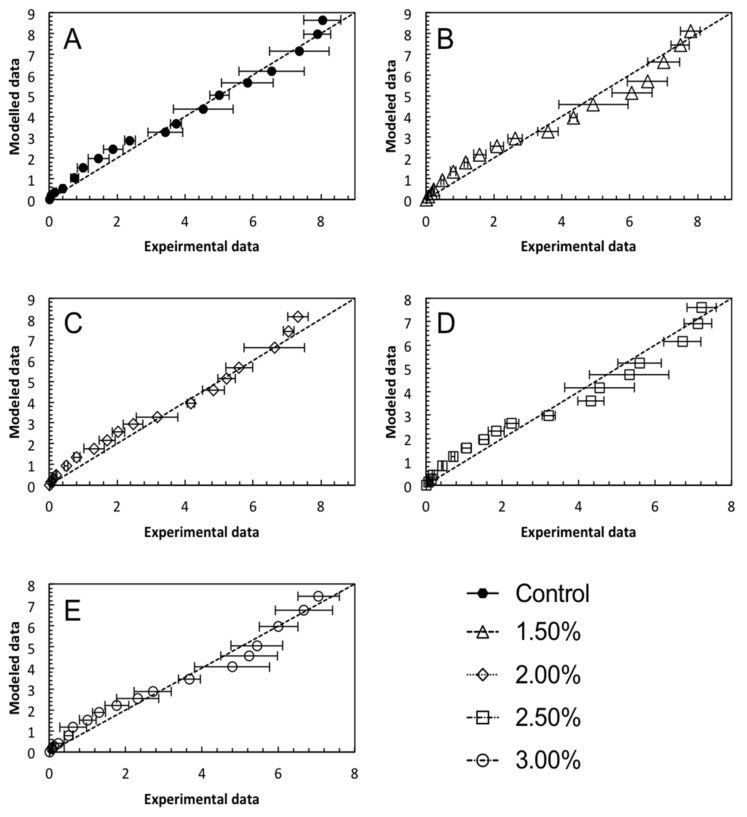
Performance evaluation plots of modelled data against experimental data of: (**A**) control bread; (**B**) 1.50% fucoidan-fortified bread; (**C**) 2.00% fucoidan-fortified bread; (**D**) 2.50% fucoidan-fortified bread; (**E**) 3.00% fucoidan-fortified bread.

**Figure 5 foods-11-00427-f005:**
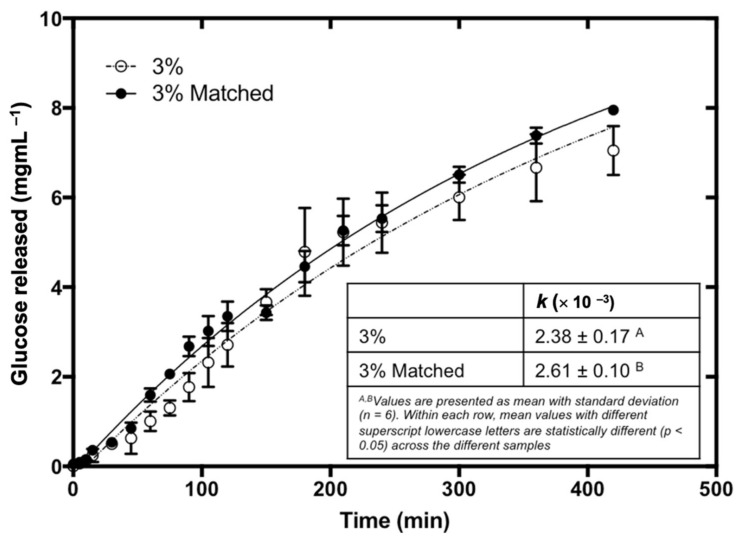
The digestion trajectories of 3% fucoidan-fortified bread and 3% matched bread, with their corresponding developed mathematical models (lines) and regressed rate constants (*k* (min ^−1^) values).

**Table 1 foods-11-00427-t001:** Mathematical modelling parameters; regressed rate constants; *k* (min^−1^); and values of TAC, pGI_bread_, pGI_glucose_, and pGL.

	Control	1.50%	2.00%	2.50%	3.00%
*k* (×10^−3^)	3.03 ± 0.23 ^a^	2.91 ± 0.18 ^a^	2.58 ± 0.23 ^b^	2.57 ± 0.12 ^b^	2.38 ± 0.17 ^b^
C_∞_ (mg/mL)	12.16 ± 0.44	12.16 ± 0.44	12.16 ± 0.44	12.16 ± 0.44	12.16 ± 0.44
R^2^	0.963	0.961	0.971	0.955	0.945
RMSE	0.322	0.457	0.363	0.413	0.383
TAC (g/5 g bread)	2.50 ± 0.36 ^a^	2.48 ± 0.31 ^a^	2.49 ± 0.36 ^a^	2.43 ± 0.34 ^a^	2.48 ± 0.38 ^a^
pGI_bread_	100 ^a^	89.2 ± 6.1 ^b^	87.3 ± 6.3 ^b^	86.7 ± 6.2 ^b^	82.3 ± 6.4 ^b^
pGI_glucose_	70 ^a^	62.5 ± 4.3 ^b^	61.1 ± 4.4 ^b^	60.7 ± 4.3 ^b^	57.6 ± 4.5 ^b^
pGL	17.5 ± 2.5 ^a^	15.4 ± 0.9 ^a,b^	15.1 ± 1.3 ^a,b^	14.6 ± 1.2 ^b^	14.1 ± 1.3 ^b^

^a,b^ Values are presented as mean with standard deviation (*n* = 6). Within each row, mean values with different superscript lowercase letters were statistically different (*p* < 0.05) across the different samples.

**Table 2 foods-11-00427-t002:** Amount of fucoidan in dialysate, digesta, and sediment; overall recovery of fucoidan; potential bioavailability; and potential bioaccessibility after in vitro digestion.

	Dialysate (mg Fucoidan/5 g Bread)	Digesta (mg Fucoidan/5 g Bread)	Sediment (mg Fucoidan/5 g Bread)	Overall Recovery (%)	F_AV_ (%)	F_AC_ (%)
Control	N.D.	N.D.	N.D.	N.D.	N.D.	N.D.
1.50%	N.D.	27.3 ± 1.4 ^a^	5.3 ± 0.6 ^a^	92.7 ± 2.4 ^a^	N.D.	77.6 ± 3.9 ^a^
2.00%	N.D.	38.1 ± 1.3 ^b^	6.7 ± 1.4 ^b^	93.8 ± 3.5 ^a^	N.D.	79.7 ± 2.7 ^a^
2.50%	N.D.	52.7 ± 4.4 ^c^	8.3 ± 1.5 ^c^	89.2 ± 5.4 ^a^	N.D.	77.1 ± 6.5 ^a^
3.00%	N.D.	61.1 ± 3.1 ^d^	9.0 ± 0.7 ^c^	91.7 ± 3.8 ^a^	N.D.	79.8 ± 4.1 ^a^

^a–d^ Values are presented as mean with standard deviation (*n* = 6). Within each row, mean values with different superscript lowercase letters were statistically different (*p* < 0.05) across the different samples. N.D. = not detected.

## Data Availability

Not applicable.

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
