# Peer review of "Fucoidan Regulates Starch Digestion: In Vitro and Mechanistic Study"

_foods, 2022, doi:10.3390/foods11030427_

Round 1

Reviewer 1 Report

I reviewed the manuscript entitled, Fucoidan regulates starch digestion: in vitro and mechanistic study. Although the manuscript attempts to address regulating the starch digestion, authors reported the low bioavailability. The direct addition of Fucoidan to dough and baking at higher temperature may affect the bioavailability of Fucoidan. I suggest authors consider microencapsulation and in vitro digestion studies that may give some interesting findings to report. Please follow other suggestions as follows   

Lines 23 and 24:  in vitro should be in Italics

Line 42: it should be "different"

Figure 1. What is the use of Figure 1? Authors purchased Fucoidan from Auckland, New Zealand. It makes sense to keep Figure 1 if the authors extract the Fucoidan from brown seaweed species.

Lines 91 and 92: the citation is not according to journal format

Lines 94 and 95: mixer company and supplier location

Line 98: baking at 200 °C for 10 min. Is fucoidan stable at higher temperatures? Authors also reported low bioavailability of fucoidan in the study. What is the purpose of fortification?

 Why did the authors not try microencapsulation in order to prevent loss of fucoidan.

At higher temperatures during Baking, most of the Fucoidan may be lost and thus authors experienced the low bioavailability.

Line 116: it must be München or Munich but not Muenchen as authors said

Sections 2.3.1. and 2.3.2 should be described to understand by the readers

Line 141: write city name in Ireland, is it Bray?

Line 145: citation format of Goñi, Garcia-Alonso, & Saura-Calixto (1997) is not correct

Line 198: Thermo Fisher Scientific Inc, please mention city and country

Line 226: post-hoc should be in Italics

Lines 233 and 234: citation format is not correct and not according to journal format

Figure 2. statistical analysis must be conducted. And, why 2.5 % showed an increased recovery of Fucoidan and 3 % decreased.

Figure 2. Where is the control group? It must be presented to show there is an increase of Fucoidan in added samples.

Lines 244 to 251: Generally, higher temperature of baking may yield loss of Fucoidan due to many factors, such as breakdown to small units and interaction with proteins. Therefore, many studies come up with microencapsulation to overcome this problem. However, authors directly fortified it into bread. There is no novelty here. Moreover, it would be better to extract Fucoidan from sources instead of procuring from commercial suppliers.    

Since the Fucoidan is fortified into bread, sensory analysis must be conducted to prove the potential application of fortified fucoidan in the functional bread industry.

Fucoidan Fortified bread may smell like seaweed and thus many vegetarian consumers may prefer not to touch this product.

None of the references are according to journal format. I suggest following journal guidelines and revising accordingly. 

Author Response

Reviewer 1:

I reviewed the manuscript entitled, Fucoidan regulates starch digestion: in vitro and mechanistic study. Although the manuscript attempts to address regulating the starch digestion, authors reported the low bioavailability. The direct addition of Fucoidan to dough and baking at higher temperature may affect the bioavailability of Fucoidan. I suggest authors consider microencapsulation and in vitro digestion studies that may give some interesting findings to report. Please follow other suggestions as follows 

First of all, we would like to thank the reviewer for carefully reading our manuscript and his/her kind comments that help us greatly to improve the quality of the manuscript

Lines 23 and 24:  in vitro should be in Italics

The words “in vitro” has been italicized.

Line 42: it should be "different"

The word has been changed to “different”.

Figure 1. What is the use of Figure 1? Authors purchased Fucoidan from Auckland, New Zealand. It makes sense to keep Figure 1 if the authors extract the Fucoidan from brown seaweed species.

The purpose of Figure 1 is to show the functional groups in the fucoidan. The fucoidan sample was not purchased from New Zealand. Instead, it was extracted by a collaborating team in New Zealand. Subsequently, the fucoidan extract was sent over and analyzed in Singapore and the structure of the fucoidan extract was determined.

Lines 91 and 92: the citation is not according to journal format

The citation has been formatted accordingly.

Lines 94 and 95: mixer company and supplier location

The mixer information has been added accordingly.

Line 98: baking at 200 °C for 10 min. Is fucoidan stable at higher temperatures? Authors also reported low bioavailability of fucoidan in the study. What is the purpose of fortification?

The purpose of this fortification is to determine if fucoidan is able to elicit its bioactivity after incorporating into a food matrix and digestion. This is one of the first studies to investigate the bioavailability of fucoidan after high temperature processing. It has also been reported previously that high molecular weight fucoidan fractions are less bioactive as compared to low molecular weight fucoidan. Hence, another purpose of this study is to elucidate the bioactivity of the fucoidan after baking with the potential of using high heat to break down the large molecular weight fucoidan fractions into lower molecular weight fractions.

Why did the authors not try microencapsulation in order to prevent loss of fucoidan.

This is one of the first tests to investigate the stability of fucoidan under high temperatures of baking. Hence microencapsulation was not considered as the loss of fucoidan had yet to be determined until this study.

At higher temperatures during Baking, most of the Fucoidan may be lost and thus authors experienced the low bioavailability.

The low bioavailability has been largely attributed to the inability of large molecular weight fucoidan fractions to pass through the dialysis tube after digestion. In fact, most of the fucoidan were retained after baking as seen in Figure 2. The loss of approximately 30% during baking has been attributed to the possibility of yeast fermenting fucoidan as additional substrate during proofing to generate additional carbon dioxide as reported in our previous study.

Line 116: it must be München or Munich but not Muenchen as authors said

It has been edited to “Munich”.

Sections 2.3.1. and 2.3.2 should be described to understand by the readers

More information has been added to Sections 2.3.1 and 2.3.2.

Line 141: write city name in Ireland, is it Bray?

City name has been included.

Line 145: citation format of Goñi, Garcia-Alonso, & Saura-Calixto (1997) is not correct

Citation has been formatted accordingly.

Line 198: Thermo Fisher Scientific Inc, please mention city and country

The city and country has been added.

Line 226: post-hoc should be in Italics

The words “post-hoc” has been italicized

Lines 233 and 234: citation format is not correct and not according to journal format

Citation format has been edited accordingly.

Figure 2. statistical analysis must be conducted. And, why 2.5 % showed an increased recovery of Fucoidan and 3 % decreased.

Statistical analysis has been conducted and added into the figure accordingly. Explanation for the difference in recovery has also been included.

Figure 2. Where is the control group? It must be presented to show there is an increase of Fucoidan in added samples.

Fucoidan in the control group was not detected. Hence the results are not presented in this figure.

Lines 244 to 251: Generally, higher temperature of baking may yield loss of Fucoidan due to many factors, such as breakdown to small units and interaction with proteins. Therefore, many studies come up with microencapsulation to overcome this problem. However, authors directly fortified it into bread. There is no novelty here. Moreover, it would be better to extract Fucoidan from sources instead of procuring from commercial suppliers.    

Since the Fucoidan is fortified into bread, sensory analysis must be conducted to prove the potential application of fortified fucoidan in the functional bread industry.

Fucoidan Fortified bread may smell like seaweed and thus many vegetarian consumers may prefer not to touch this product.

This is one of the first studies to investigate the use of fucoidan as a functional ingredient in bread matrix. Hence the use of microencapsulation is not yet considered. Sensory evaluation has been conducted and will be reported as part of a separate manuscript.

None of the references are according to journal format. I suggest following journal guidelines and revising accordingly. 

The references have been edited accordingly.

Reviewer 2 Report

This study (foods-1539277) investigates an ingredient that can decrease the GI value of bread.

Comments are as below:

  1. Please add some sentences of the previous study (Koh H S A, Lim S E V, Lu J, et al. Bioactivity enhancement of fucoidan through complexing with bread matrix and baking. LWT, 2020, 130: 109646) on the texture improvement of bread by adding fucoidan.
  2. Please measure the Mw of fucoidan before and after baking, as this can be the novelty of this study.
  3. line 59-60: It was hypothesized that the incorporation of fucoidan into bread formula would reduce 59 the in vitro starch digestibility and glycaemic potential of baked bread. This has been proved in previous studies. What is the research question of this study, please clarify.
  4. Line 284-301, if it shall be in the introduction part?
  5. How about the hypothesis rose in the previous paper: ‘complexing with bread matrix’? This could happen in this study as well, and it is not justified.

Author Response

Reviewer 2:

This study (foods-1539277) investigates an ingredient that can decrease the GI value of bread.

Comments are as below:

Please add some sentences of the previous study (Koh H S A, Lim S E V, Lu J, et al. Bioactivity enhancement of fucoidan through complexing with bread matrix and baking. LWT, 2020, 130: 109646) on the texture improvement of bread by adding fucoidan.

The texture improvement of bread upon addition of fucoidan has been elaborated briefly in Lines 50-53.

Please measure the Mw of fucoidan before and after baking, as this can be the novelty of this study.

Due to the low dosage of fucoidan added into bread, we were unable to recover sufficient quantities of fucoidan for Mw determination after baking.

line 59-60: It was hypothesized that the incorporation of fucoidan into bread formula would reduce 59 the in vitro starch digestibility and glycaemic potential of baked bread. This has been proved in previous studies. What is the research question of this study, please clarify.

This is one of the first studies to investigate the potential of fucoidan in reducing in vitro starch digestibility and glycaemic potential in a food system. Previous studies have only reported the ability of fucoidan to inhibit starch digesting enzyme but there have been no reports on the ability of fucoidan to reduce glycaemic potential in a food matrix.

Line 284-301, if it shall be in the introduction part?

Lines 284-301 describes the proposed mechanism by which fucoidan may have inhibited starch digesting enzymes and hence was used in the discussion and not the introduction part.

How about the hypothesis rose in the previous paper: ‘complexing with bread matrix’? This could happen in this study as well, and it is not justified.

It is possible that the bioactivity of fucoidan is enhanced through complexing with the bread matrix and that was described in our previous paper. This manuscript focused on investigating the potential of fucoidan to reduce glycaemic potential and in vitro starch digestibility in a food matrix (i.e., bread).

Reviewer 3 Report

In recent years, there has been an increasing trend towards products with a low glycemic index. In this respect, I think the subject is interesting. The article nevertheless needs some revisions.

The originality of this work should be clearly written in the introduction section.

Line 102 was the extraction and quantification carried out for baked breads? It should be clearly described

Line 230 the words generally should be removed.

Line 232 –it should be 18-30 %. 

Figure 3 needs to be revised in order to increase its visually. The numbers on the Y-axis must be rearranged. It is very difficult to distinguish the samples from each other.

Do the results of the article provide an innovative perspective for the addition of fucoidan for bread with a lowered glycemic index? I think this should be stated in both the abstract and the conclusion.

Author Response

Reviewer 3:

In recent years, there has been an increasing trend towards products with a low glycemic index. In this respect, I think the subject is interesting. The article nevertheless needs some revisions.

The originality of this work should be clearly written in the introduction section.

The originality of this work has been added into the introduction.

Line 102 was the extraction and quantification carried out for baked breads? It should be clearly described

Yes, the extraction and quantification were carried out for baked breads, and this information has been added in the manuscript.

Line 230 the words generally should be removed.

The word “generally” has been removed.

Line 232 –it should be 18-30 %. 

The mistake has been corrected.

Figure 3 needs to be revised in order to increase its visually. The numbers on the Y-axis must be rearranged. It is very difficult to distinguish the samples from each other.

Due to the standard deviation at each time point, it is difficult to improve the distinctions between the samples. However, if one ignore the data points and look at the overall trend lines, it is easier to distinguish the samples via the different dotted lines. The information on digestibility and rates of digestion has also been tabulated in Table 1.

Do the results of the article provide an innovative perspective for the addition of fucoidan for bread with a lowered glycemic index? I think this should be stated in both the abstract and the conclusion.

The relevant information has been added following your kind suggestion.

Reviewer 4 Report

Comments are follows.

・The authors use incorrect statistical methods.

  1. The authors described ‘All analyses were carried out in triplicates, with two or three repeats in each replicate’ (L222). This means that all analyses were performed with n=3 (Repeated measurement of same sample do not increase the number). However, n=6 is described in the statistical analysis in Table 1, Table 2, and Figure 5.
  2. The authors use Duncan’s multiple range test in statistical analysis. However, Duncan’s multiple range test have a greater risk of making type 1 errors. Therefore, the authors should examine statistical analysis using another statistical test, such as Tukey’s test, again.
  3. The authors did not statistical analysis in ‘In-vitro starch digestibility of bread’ (Fig. 3), however, the authors described that ‘At the end of 105min of digestion, the concentration of RS released by the fucoidan-fortified bread was 23 – 34% significantly lower than that by the control baked bread. This suggests that fucoidan was able to significantly reduce starch digestion in bread as early as 105 min in the intestinal phase.’ If the authors want to use the word ‘significantly’, statistical analysis of the data in Fig. 3 should be performed (‘significantly’ is used in other sentences).

・Although the authors cite reference [9], the authors should investigate the inhibitory ability of fucoidan used in this study againstα-amylase, α-glucosidase, and amyloglucosidase.

・As the authors describe in their manuscript, alpha-amylase and alpha-glucosidase inhibition by fucoidan has been well examined including the report by the author’s research group.

・In Table 1, fucoidan-supplemented bread significantly decreased pGI and pGL compared to the control bread sample, however the significance was not observed between the 1.5% sample and the 3.0% sample. The authors should explain this matter.

Author Response

Reviewer 4:

The authors use incorrect statistical methods.

The authors described ‘All analyses were carried out in triplicates, with two or three repeats in each replicate’ (L222). This means that all analyses were performed with n=3 (Repeated measurement of same sample do not increase the number). However, n=6 is described in the statistical analysis in Table 1, Table 2, and Figure 5.

It was reported as n=6 and not n=3 because within each replicate processing, 2 different bread buns were used and analyzed separately. That is, for each replicate processing, 2 bread buns were used independently to extract and quantify fucoidan, as well as to undergo in vitro digestion to determine the digestibility and glycaemic potential of the bread. Hence, it was not the case of “repeated measurement of the same sample” as different bread samples were used within each of the three replicate processing, i.e., a total of 6 samples under every processing condition.

The authors use Duncan’s multiple range test in statistical analysis. However, Duncan’s multiple range test have a greater risk of making type 1 errors. Therefore, the authors should examine statistical analysis using another statistical test, such as Tukey’s test, again.

Thanks for your insightful comment and suggestion. The results have been re-analyzed by Tukey’s test as suggested.

The authors did not statistical analysis in ‘In-vitro starch digestibility of bread’ (Fig. 3), however, the authors described that ‘At the end of 105min of digestion, the concentration of RS released by the fucoidan-fortified bread was 23 – 34% significantly lower than that by the control baked bread. This suggests that fucoidan was able to significantly reduce starch digestion in bread as early as 105 min in the intestinal phase.’ If the authors want to use the word ‘significantly’, statistical analysis of the data in Fig. 3 should be performed (‘significantly’ is used in other sentences).

We would like to confirm that statistical analysis was performed at each time point throughout the digestion process. When the word 'significant'/'significantly' is used in the text, it indeed means 'statistically significant'. However, due to crowding of the figure, the statistical results were not able to be indicated in the diagram.

Although the authors cite reference [9], the authors should investigate the inhibitory ability of fucoidan used in this study against α-amylase, α-glucosidase, and amyloglucosidase.

As the authors describe in their manuscript, alpha-amylase and alpha-glucosidase inhibition by fucoidan has been well examined including the report by the author’s research group.

The fucoidan used in this study has been reported to be inhibitory against α-amylase, α-glucosidase, and amyloglucosidase in our previous study (Koh et al., 2020).

Koh, H.S.A., J. Lu, and W. Zhou, Structural Dependence of Sulfated Polysaccharide for Diabetes Management: Fucoidan From Undaria pinnatifida Inhibiting α-Glucosidase More Strongly Than α-Amylase and Amyloglucosidase. Frontiers in Pharmacology, 2020. 11: p. 831.

The novelty in this study is in the fact that this is one of the first studies to investigate the ability of fucoidan to lower the starch digestibility and glycaemic potential in a food system, i.e., bread.

In Table 1, fucoidan-supplemented bread significantly decreased pGI and pGL compared to the control bread sample, however the significance was not observed between the 1.5% sample and the 3.0% sample. The authors should explain this matter.

Using a linear regression correlation test, a significant linear correlation was observed which suggests a dose-dependent reduction in pGI and pGL with fucoidan concentration (Lines 372 – 373). Therefore, the lack of significance between the 1.5% and the 3.0% samples despite of the presence of a significant linear correlation could be attributed to the relatively large standard deviations of experimental data points.

Round 2

Reviewer 1 Report

Authors failed to address the comments raised by me. The direct addition of Fucoidan to dough and baking at higher temperature yielded the low bioavailability of Fucoidan. Thus, performing
microencapsulation and in vitro digestion studies may give some interesting findings to report.

The experimental approach is not correct.Authors reported the lo

Author Response

First of all, thank you very much for kindly making further comments.

Please allow us to gently point out the difference between bioavailability and bioaccessibility. It is the bioaccessibility that reflects the amount of fucoidan surviving the baking process and therefore helps to conclude whether adopting microencapsulation becomes essential.

From our results, while the overall bioavailability of fucoidan was low, the bioaccessibility of fucoidan was relatively high (Table 2: 77-79%).

Bioavailability in this paper refers to the amount of fucoidan that passed through the dialysis tubing (which mimics the intestinal wall) and therefore indicated how readily it would be absorbed via the blood stream. The low bioavailability can be attributed to the large molecular weight of fucoidan fragments after baking, despite that their sizes were already reduced from the originally fortified fucoidan, and thus they were unable to pass through the dialysis tubing.

On the other hand, bioaccessibility refers to the amount of fucoidan that can be potentially absorbed in the intestinal lumen and includes the amount of fucoidan remained in the digesta (intestinal chyme). The relatively high bioaccessibility of fucoidan at 77-79% indicates that most of the fortified fucoidan were retained in the digesta, which might be absorbed via means other than passive diffusion. This suggests that most fucoidan were not destroyed under the high baking temperatures and hence improvement from microencapsulation, if conducted, would be limited. 

As shown in Figure 2, the overall recovery of fucoidan after baking (in the bread sample) was relatively high at 70-82% (Figure 2), indicating limited damage due to baking. While we are not against conducting microencapsulation to further reduce the limited loss caused by baking, we hope you would agree that this does not immediately become essential and it won’t reduce the significance of the current study.

We earlier demonstrated some benefits (e.g., bioactivity) of fucoidan comlexing with bread matrix as well as through baking (Koh et al., 2020). Microencapsulation of fucoidan is likely to reduce those benefits. Nevertheless, we will conduct microencapsulation of fuciodan and report the results in a separate manuscript in future. 

Koh, H.S.A.; Lim, S.E.V.; Lu, J.; Zhou, W. Bioactivity enhancement of fucoidan through complexing with bread matrix and baking. LWT, 2020, 130, 109646.

Reviewer 2 Report

manuscript is well revised

Author Response

Our sincere thanks to the reviewer for her/his kind recommendation.

Reviewer 4 Report

Although the authors replied as ‘The results have been re-analyzed by Tukey’s test as suggested’; however, ‘a post-hoc analysis was carried out using Duncan’s multiple range test and Tukey’s range test (L235-236)’ is written in the revised manuscript.

Author Response

We would like to apologize for the confusion by leaving both Duncan's multiple range test and Tukey's range test in the revised manuscript. We did take the reviewer's earlier kind suggestion on re-analyzing the data by Tukey's test and the results of the Tukey's test were the same as the previous outcome of Duncan's test. That's why we thought perhaps it's better to state both tests in Section 2.9.

To avoid this confusion, ‘Duncan’s multiple range test and’ has been removed from Section 2.9 (L235 – 236).